# Estimating rates of treatment delay for malaria fevers among children in Sub-Saharan Africa 2006–2022

Jailos Lubinda [1,2,6] ✉, Susan F. Rumisha[1,3,4,5,6], Paulina Dzianach[1], Michael McPhail[1], Adam Saddler[1], Annie Browne[1], Francesca Sanna[1], Yalemzewod Gelaw[1,3], Paul Castle[1], Juniper B. Kiss[1], Joseph Harris[1], Jennifer A. Rozier[1], Camilo Vargas[1], Punam Amratia[1,4], Tasmin L. Symons[1,3], Ewan Cameron[1,3], Peter W. Gething[1,3,7] & Daniel J. Weiss[1,3,7]

Late diagnosis and treatment of malaria increase the odds of severe disease by nearly 2.8 times, enhance transmission rates, compromise drug effectiveness, and trigger malaria outbreaks. No continent-wide estimates for malaria treatment delay exist. We estimate delay rates among African children treated for malaria between 2006 and 2022, using 177 nationally representative surveys. We found that 60% [95% UI 45.7–72.8] of treated children experienced >24 h of delay, while 29% [95% UI 18.9–41.2] faced delays exceeding 48 h, affecting 33 million and 16 million children, respectively. Spatiotemporal variability exists across Africa. Somalia has the highest (76% [95% UI 39.7–97.9]) and Tanzania has the lowest (35.3% [95% UI 11.5–53.8]) delay rates. Overall, initial improvements in treatment delay stagnated post-2015, but East Africa showed the most progress, while Central and West Africa experienced increases. Socioeconomic factors and residence influenced delays, with poorer and rural populations facing higher rates. These findings are vital for policymakers to enhance malaria case management, access to effective treatment, and reduce malaria mortality among children.

Since the African Summit on Roll Back Malaria's Abuja Declaration[1], there has been a renewed global emphasis on the importance of early diagnosis and treatment <24 h of the onset of symptoms, which is the optimal strategy for managing uncomplicated malaria[2]. Prompt access to care, diagnosis, and effective antimalarial treatment is crucial for malaria case management and is vital to malaria control and elimination strategies[3,4]. Evidence shows that treatment with a >24-h delay significantly increases the odds of severe malarial phenotypes by 33%, is associated with odds ratios of 2.79 for severe malarial anaemia in children delayed by 2–3 days, 5.46 for a delay of >7 days[5], and can perpetuate or trigger malaria outbreaks in low-transmission settings

(<10% P. falciparum parasite rate [$PfPR_{2-10}$])[6]. Prompt treatment <24 h is reported to potentially avert nearly half of these severe cases in both children and adults[5], highlighting the critical importance of early intervention. For a host of reasons, however, many febrile children in Africa experience delays in reaching the care and receiving the treatment they need.

The time between fever onset and receiving treatment is crucial because it directly affects the probability of disease progression to severe illness or death[5,7]. Delayed treatment increases the likelihood that patients will develop long-term consequences of severe diseases, which are significantly associated with post-discharge mortality[8]. For

[1]The Kids Research Institute Australia, Perth Children's Hospital, Nedlands, WA, Australia. [2]Macha Research Trust, Choma, Zambia. [3]Curtin University, Bentley, WA, Australia. [4]Ifakara Health Institute, Dar es Salaam, Tanzania. [5]National Institute for Medical Research, Dar es Salaam, Tanzania. [6]These authors contributed equally: Jailos Lubinda, Susan F. Rumisha. [7]These authors jointly supervised this work: Peter W. Gething, Daniel J. Weiss. ✉e-mail: Jailos.Lubinda@thekids.org.au

example, children with severe conditions such as cerebral malaria are also associated with substantial neurodevelopmental sequelae like stroke, cognitive impairment, and an increased risk of epilepsy[7]. Other potential consequences include increases in malaria reservoirs for onward transmission to mosquitoes[9], leading to prolonged parasite carriage, which increases the risk of drug resistance due to suboptimal parasite clearance and compromise of antimalarial drug effectiveness in routine healthcare settings[10–12]. Findings from the literature show that delayed treatment by >24 h since fever onset increases onward transmission rates by up to 50%, and the mean infectivity of individuals with prolonged delay or untreated infections surpasses that of infections treated <24 h by 29–51 times[9].

Recent multisite studies on malaria mortality across Kenya, Mali, Mozambique, and Sierra Leone show how delayed treatment remains highly prevalent and consequential, reporting that over 62% of malaria deaths in under-5 children had received an antimalarial before death, but too late for it to be effective[13]. In Angola, another study showed that 96% of malaria patients who died <24 h of admission to the health facility received appropriate care but did not survive because they had severe disease on arrival, possibly caused by delay[14]. Delayed antimalarial treatment, therefore, remains a serious threat to child survival in many African countries.

Nearly half a million children aged under 5 in Africa succumb to malaria each year, representing more deaths than all other age groups combined[15]. Children in Africa have an increased risk of severe malaria (especially severe anaemia), and death due to their less-developed malaria immunity, and progression to severe outcomes can be rapid[16,17]. In the event of no or delayed treatment, there is a rapid progression of the disease due to an exponential increase in parasites[7], particularly in *P. falciparum*. All this can occur within a few hours of symptom onset[16], highlighting the critical importance of prompt treatment, especially in children, as studies have also shown that most malaria deaths among young children occur within 2–3 days after the onset of symptoms[18].

Despite the importance of delayed treatment in driving preventable deaths from malaria, there is a continent-wide knowledge gap in understanding the extent of delayed treatment, trends over time, and how these vary geographically. Using data from 177 nationally representative demographic health, and multiple indicator cluster surveys conducted between 2006 and 2022 across sub-Saharan Africa (SSA), this analysis modelled data from 111,325 under-5 febrile children who received antimalarial treatment in endemic Africa. We address the national, regional, and continental information gap by presenting the first systematic attempt to evaluate levels of delayed treatment across the continent. The fraction of treatment delay was defined as delay when >24 h, moderate (24 to <48 h) and severe (>48 h), and estimated using a hierarchical Bayesian model with spatiotemporal random effects. The impact of demographic, socioeconomic, and health system covariates was assessed using the same model.

## Results

### Spatial patterns of treatment delay

We estimated that, in 2022, 60.0% [95% UI 45.7–72.8] of all children under 5 who received antimalarial treatment in Africa experienced delayed treatment, while 28.8% [95% UI 18.9–41.2] were severely delayed. The latter fraction represents over 16 million [95% UI 10,811,444–32,714,816] children treated for malaria receiving their treatment outside of WHO-recommended timelines for prompt and effective treatment. According to the most recent edition of the WHO Guidelines for treating malaria, published in 2015, prompt and effective treatment must be within 24–48 h of the onset of malaria symptoms[16]. Across the continent, national-level fractions of treatment delay ranged from 35.3% [95% UI 11.5–53.8] in Tanzania to 76% [95% UI 39.7–97.9] in Somalia (Fig. 1a, b, and Supplementary Data 1), while severe delay ranged from 49.1% [95% UI 25.3–93.2] in Somalia to 14.7% [95% UI 3.1–41.2] in Gambia (Fig. 1b, SI Results Table S2).

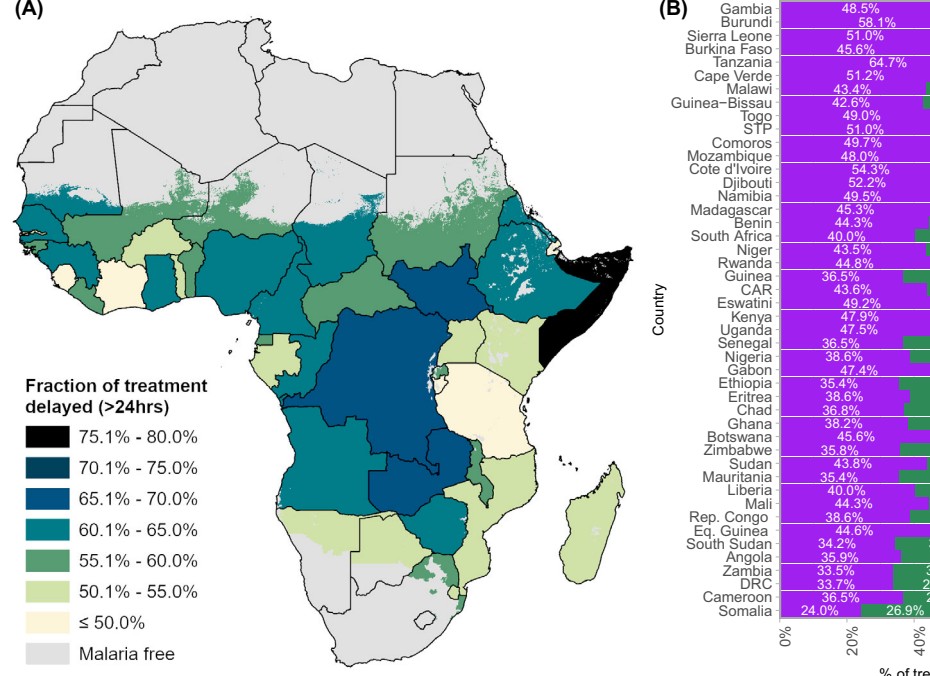

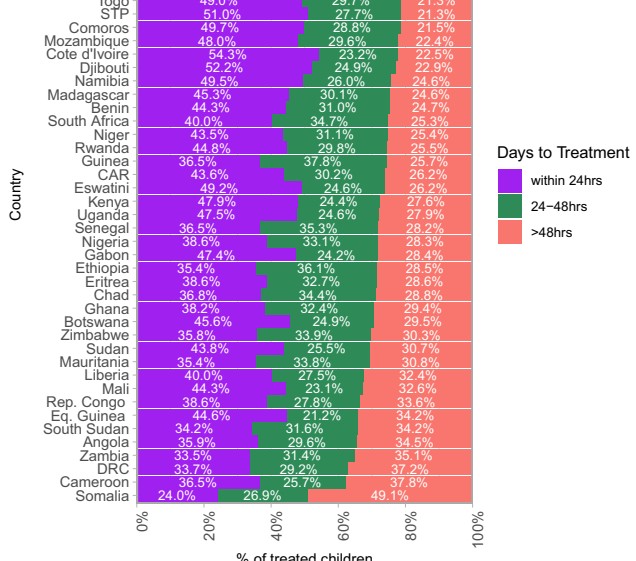

**Fig. 1 | Spatial patterns distribution of delayed treatment in 2022. A** Spatial patterns of estimated fraction of malaria treatments delayed (>24 h of symptom onset) in 2022 Darker colours represent countries of higher delay, while lighter colours represent lower levels of delay. **B** Estimated fraction of treatments for malaria by treatment timing, by country, for 2022. Purple colour represents proportion receiving treatment within 24 h (prompt treatment), green colour represents proportion of moderate delay, and red colour represents proportion of severe delay. DRC (Democratic Republic of Congo), CAR (Central African Republic), STP (Sao Tome and Principe), Rep. Congo (Republic of Congo), Eq. Guinea (Equatorial Guinea).

## Trends in treatment delay between 2006 and 2022

We estimate that the continent-wide fraction of malaria treatment delay fell from a peak of 71.7% (95% UI 60.5–80.8) in 2008 to 60% [95% UI 45.7–72.8] in 2022 (Fig. 2a). However, the majority of this decline occurred prior to 2015, after which progress has been modest. The figure also shows the fraction of treatment delay of subregions relative to the mean continental fraction and trends. The fraction of delay for West Africa (AFRO-W) was constantly below the continental mean, Eastern Mediterranean (EMRO) was above it, while the other three crossed it for better or worse. Figure 2b shows the estimated trends at subregional levels, with their 95% UIs. The WHO East Africa (AFRO-E) subregion experienced the greatest and most consistent improvement, with a 22% decline in the fraction of delay between 2006 and 2022. In contrast, Southern Africa (AFRO-S) and the EMRO saw marginal declines of 6% and 5%, respectively. The Central Africa (AFRO-C) and AFRO-W subregions observed increases in the fraction of delay by 10% and 4%, respectively. Broadly, AFRO-W and AFRO-S reflect the continental trend of an initial decline followed by a stall, resulting in only marginal compound reductions in the fraction of delay. While the trend for EMRO is surprisingly flat, central, and west Africa are the most concerning, with delayed treatment having become more common by 2022 versus 2006. Overall, 85% (39/46) of the individual countries made some progress over the study period, with declines ranging from 1% to 27%, while 15% (7/46) experienced greater rates of treatment delay in 2022 compared to 2006 (see SI Fig. S8). The period between 2010 and 2015, saw the greatest improvements in the provision of prompt antimalarial treatment and experienced the most reduction in delayed treatment recording declines in 82% of the countries (39/46) and by 9.3% (95% UI 7.4–10.5) continent-wide.

## Trends-missed opportunity

Among febrile children treated in 2022, an estimated 41% [95% UI 27.2–54.3] received antimalarials promptly, about one-third had a moderate delay (30% [95% UI 14.0–35.4]), and another nearly one-third were severely delayed (29% [95% UI 18.9–41.2]). In Fig. 3 and SI Fig. S9, we present an illustration of the fraction of treated cases delayed among febrile children receiving antimalarials, focusing on the differences between the fraction of moderate and severe delay. The figure provides an overview of the geographical distribution and concentration of under-5 malaria cases by delay, emphasising the opportunity for improvement in addressing treatment delay within different subregions. Although the fraction of delay may be similar (SI Fig. S10), the concentration of treated cases delayed is highest in West Africa (several-fold higher than the others), followed by Central Africa, East Africa, Southern Africa, and the East Mediterranean (Fig. 3, SI Fig. S11a). Central Africa also shows a high number of cases, but with a more balanced distribution among the two categories of delay. East Africa and Southern Africa demonstrate a small fraction of the overall number of cases delayed compared to West and Central Africa. However, only East Africa made notable progress in reducing delayed cases across both moderate and severe delays over time. Southern Africa shows a relatively stable trend with a slight corresponding decrease in moderately delayed cases.

Given the subregional distribution of treated malaria cases, deaths and fraction of delay, certain countries emerge as dominant contributors across all metrics. In West Africa, Nigeria stands out with a significant share of 55% of the subregion's cases, 61% [95% UI 25.8–84] delay, and 28% [95% UI 7.1–54.3] severe delay. Similarly, Central Africa is dominated by the DRC with a high subregional burden (54%), 66% [95% UI 32.8–94.1] delay, and 37% [95% UI 32.8–94.1] severe delay. Similarly, these countries carry disproportionate shares of delayed cases within the continent and more than half of the delay burden in their subregions (SI Fig. S11b-c). Angola and Cameroon both have over 60% treatment delay and at least one-third is severely delayed. Contrastingly, in East and Southern Africa, although Uganda, Mozambique,

and Tanzania are prominent, contributing 23%, 19%, and 14% of cases, their ranking in moderate and severe delay fractions is lower (Supplementary Data 1 and SI Table S5).

Overall, 79% [95% UI 54.6–100] of the burden of severely delayed children was carried by just 11 countries (Angola, Burkina Faso, Cameroon, the Democratic Republic of the Congo, Côte d'Ivoire, Mali, Mozambique, Niger, Nigeria, Uganda and Tanzania), which are also responsible for about 70% of the global malaria incidence, 73% of global malaria deaths[19]. Plots and tabular level outputs at the country level are available in SI Fig. S12 and Table S5. Free access to generated estimates for all countries, subregions and continent under the full study period is accessible at https://doi.org/10.5281/zenodo.17096726.

## Differential trends in treatment delay by urban–rural and wealth gradients

Continental and subregional trends in delayed treatment in rural versus urban areas and among the households from the lower two versus the upper two wealth quintiles are shown in Fig. 4 and SI Fig. S13. During the study period, we observed that overall, children from poorer households and administrative areas progressed and experienced much greater declines in the fraction of treatment delays than those in higher quintile households (SI Fig. S13). A similar trend is observed between rural and urban populations, where the fraction of rural-residing children delayed in receiving treatment has declined faster than their urban counterparts. In some instances, especially in more recent years, we estimated that children receiving antimalarials in rural areas were equally likely to receive prompt treatment as those in urban areas. In contrast, children from higher wealth quintile households experienced minimal progress on average and, in some instances, an increasing fraction of treatment delay. Although the subregional trends across the two strata of wealth and location are consistent with the continental trends, in 2022, East Africa and Southern Africa exhibit at least a 5% higher fraction of treatment delay and a continued decline in delay among their rural populations compared to their urban and wealthier counterparts.

## Discussion

Tracking improvements in access to care for febrile illness and malaria in Africa requires understanding the extent to which antimalarial treatment is received promptly or otherwise. The data assembly and statistical modeling study presented here represents the first attempt to comprehensively characterise the spatial patterns, temporal trends, and key strata of delayed malaria treatment for children under five years across the continent over recent decades. This analysis sheds light on a critical but relatively underappreciated factor that likely contributes to sub-Saharan Africa's disproportionate share of severe malaria and malaria mortality.

Our findings reveal that over 60% of children treated for malaria face an elevated risk of disease progression to severe malaria or death due to delayed treatment, with half of these cases being severely delayed. This fraction translates to millions of children that, despite ultimately receiving an antimalarial, are denied the benefit of prompt, effective treatment. The magnitude of this challenge highlights the importance of addressing delayed treatment to reach the WHO global target of reducing malaria mortality rates by at least 90% by 2030[20].

The trends show an increase in treatment delays from 2006 to 2008, a decline until 2015, followed by a plateau until 2022. These trends mirror the stalled progress against malaria disease burden in Africa post-2015, and plausibly, the absence of marked improvement in promptness of care over the same timeframe may be one contributory factor. Quantifying the factors responsible for trends in delayed malaria treatment is beyond the scope of this research, but consideration of concurrent trends in the broader global effort against the disease provides useful context. Foremost among these is the massive scale-up of international funding and scientific focus devoted to

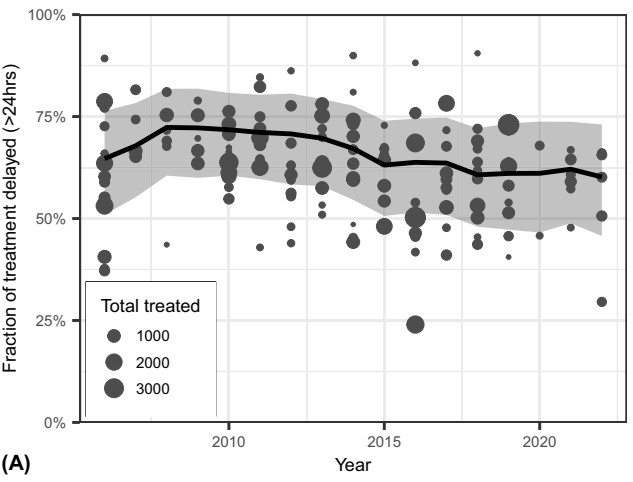

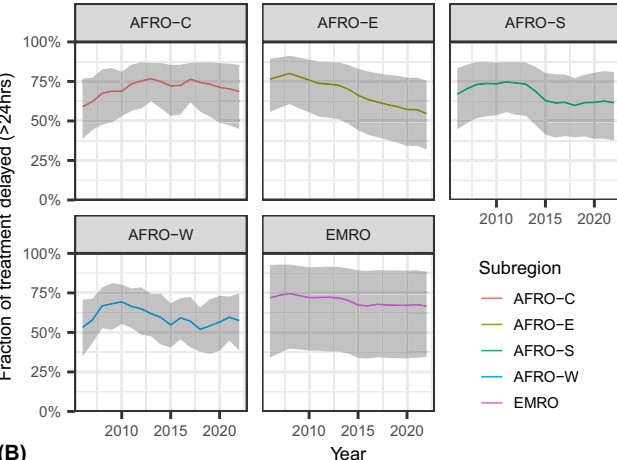

**Fig. 2 | Continental and sub-regional trends of delayed treatment. A** Estimated population-weighted continental trends of the fraction of treatment delay and the relative subregion means. Each dot represents a survey included in the model, with the size corresponding to the number of child respondents reporting antimalarial treatment and the vertical position indicating the observed fraction of delayed treatment. The black line is the posterior mean estimate; the grey envelope is a 95% uncertainty interval, and the coloured lines represent subregions. **B** Estimated population-weighted subregional trends. Coloured lines are posterior mean estimates; grey envelopes are 95% uncertainty intervals. WHO subregional groupings: AFRO-C (Central Africa); AFRO-E (East Africa); AFRO-S (Southern Africa); AFRO-W (West Africa); EMRO (Eastern Mediterranean Region).

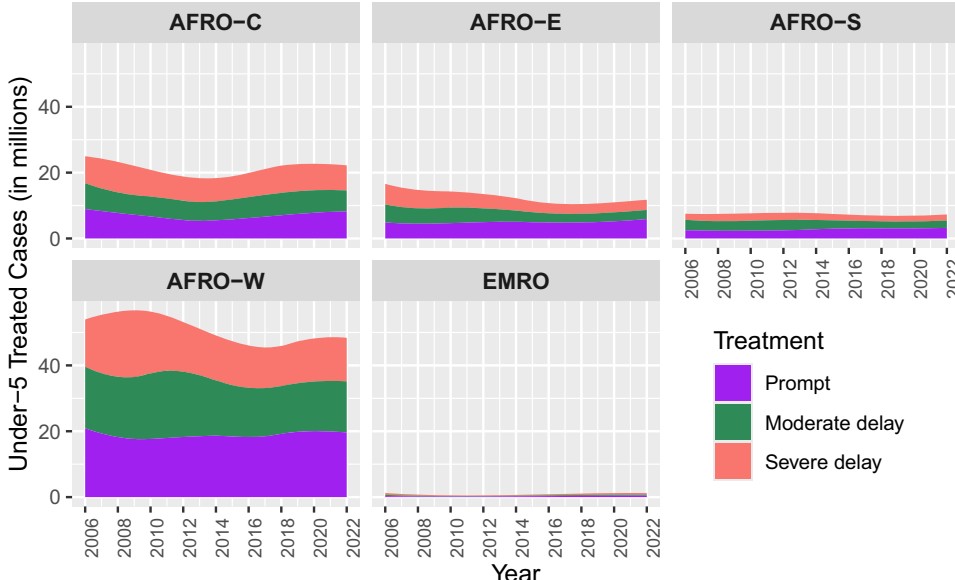

**Fig. 3 | Subregional trends of under-5 treated cases by treatment delay (moderate delay and severe delay).** The y-axis shows the number of children treated with antimalarials in millions.

malaria leading up to 2010 and 2015, concurrent with heightened country-level action inspired by the Millennium Development Goals (MDG). During this period, investments targeted socio-economic development to uplift the marginalised and mostly rural populations and countries furthest from the goals, as well as efforts to expand access to core interventions for malaria prevention, diagnosis, and treatment as part of universal health coverage towards the MDG health goal[21]. This period of increased attention and funding was concurrent with the greatest improvements in the provision of prompt antimalarial treatment within SSA especially between 2010 and 2015.

The study also demonstrates significant overall progress in regional trends across most regions. The notable increase in prompt treatment and the reduction in severe delay in AFRO-E, AFRO-S and EMRO subregions signifiy a promising trajectory towards ensuring more children receive timely care. The faster decline of delayed treatment in rural and low SES households of East Africa compared to urban areas or high SES and other regions may reflect a prevailing complex interplay of healthcare access and socio-demographic factors. Expanding urban slums in East Africa face worsening health conditions due to rapid urbanisation, overcrowding, and inadequate healthcare, leading to a slower decline in health outcomes and reversing the 'urban advantage' traditionally associated with better access to services[22,23]. In contrast, rural areas have significantly benefited from targeted health interventions, community-based care, and improved outreach, resulting in better care-seeking behaviour and reduced delays[24,25]. In countries such as Kenya, Tanzania, Ethiopia, and Uganda, neonatal mortality rates have also declined faster in rural settings, narrowing the urban-rural gap and highlighting poor infrastructure and limited access to care as key drivers of slower progress in urban areas[22–24,26].

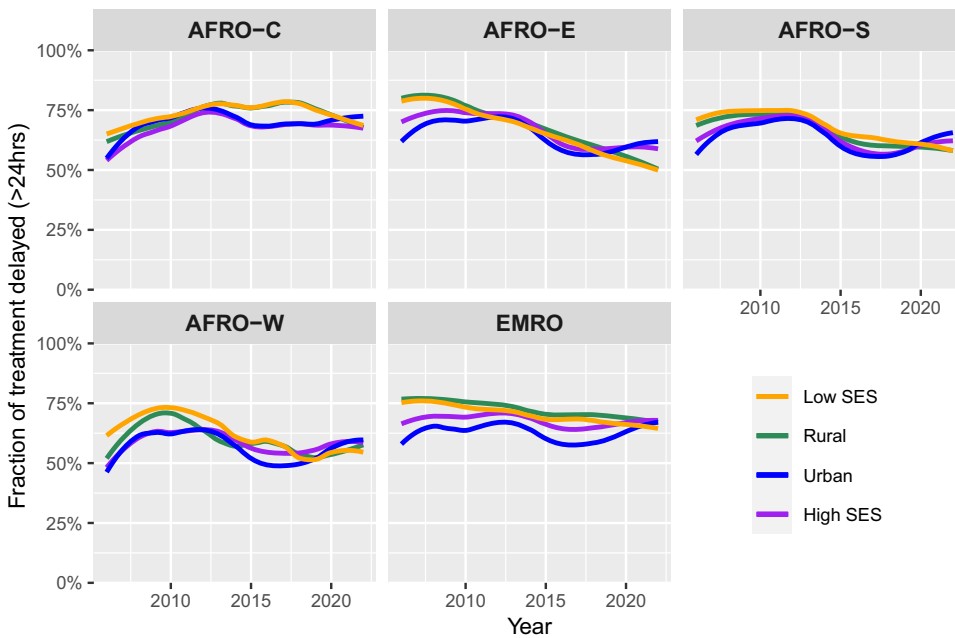

**Fig. 4 | Subregional trends across urban, rural, high SES, and low SES households.** Subregions here represent WHO subregional groupings: AFRO-C (Central Africa); AFRO-E (East Africa); AFRO-S (Southern Africa); AFRO-W (West Africa); EMRO (Eastern Mediterranean Region); SES (Socioeconomic status).

Further, the decrease in moderate delays across most regions is a positive development, but sustained efforts are required to continue this trend and ensure timely treatment for all patients. These improvements translate into potential reductions in case fatality rates and highlight overall enhancements in health systems. Understanding the distribution of treatment delays across the transmission spectrum is crucial for identifying bottlenecks and further developing tailored strategies for reducing delays. Targeting regions experiencing significant delays can potentially enhance health outcomes on a broader scale.

Despite this overall progress, significant regional disparities persist. In 2022, West and Central Africa remained the most affected by treatment delays, with increasing trends in severe delays and declining prompt treatment, while also contributing the highest under-5 malaria mortality ratios[19]. For instance, under-5 children in West Africa (excluding Nigeria and Mauritania) accounted for 37% of total cases in the region but 73% of the mortality, while in Central Africa, it was 43.8% of total cases but 60% of the deaths. In contrast, children in East and Southern Africa collectively accounted for only 35.1% of cases and 47% of deaths[19]. It is concerning that West and Central Africa have the highest concentrations of delayed treatments and disproportionately higher mortality ratios among under-5 children. This is likely due to the higher population densities in these regions, which naturally lead to higher absolute numbers of cases. Although regions with higher malaria burdens may naturally exhibit higher absolute numbers of delayed treatments due to varying levels of malaria prevalence and healthcare access, this signifies a greater opportunity for health systems intervention to address bottlenecks to prompt treatment. It underscores the importance of targeted public health interventions to optimise treatment outcomes and address worsening delays among vulnerable populations and regions.

At a country level, the concentration of delay and mortality in high-burden countries like Nigeria, DRC, Angola and Cameroon is not surprising because it reflects the magnitude of the issue. However, contextualising these figures within each region's unique healthcare landscape is still essential. For instance, regions with higher malaria burdens naturally exhibit higher numbers of delayed treatments and mortality, underscoring the urgency for responsive interventions to improve promptness. Conversely, regions with lower case burdens but

higher delay rates may suffer from higher case fatality magnitudes due to severe disease driven by low immunity to malaria. Therefore, it is crucial to address these gaps, especially in regions like West and Central Africa, where delays in treatment are more prevalent, to reduce the overall burden of malaria and improve treatment outcomes. Additionally, countries with low burdens but high delays also require attention. The heterogeneity in the disproportionate fraction of delay in specific regions and countries suggests that tailored strategies could optimise targeted treatment interventions, significantly reducing the impact of delay for malaria-related morbidity and mortality among these young and vulnerable populations.

Our study highlights that millions of children are denied the benefit of prompt treatment for malaria, increasing their risk of severe disease and death. Previous research has reported that over one-third of children with a fever do not seek care[20,27]. Our findings further indicate that only around one-quarter of those treated with an antimalarial in SSA receive prompt treatment. This has huge implications for achieving WHO's global target of reducing mortality by 90% by 2030 and improving prompt access to health care by reducing treatment delays should be prioritised by both malaria-focused and global health funders. Current guidelines emphasising treatment 24 to <48 h must be actively implemented, focusing on treatment <24 h.

With the evidence already provided that malaria treatment within 24 h of symptom onset improves treatment outcomes and reduces risk of death among children[5,7,9–12], the focus must prioritise strategies that ensure local health policy implementers actively pursue prompt access to effective treatment as soon as possible, especially <24 h[10,28]. Constant behavioural change communication and prompt treatment messaging, encouraging communities, particularly mothers, to seek care promptly, especially for fever could further raise awareness on the dangers of delayed treatment, promote early intervention and help close this gap[28–30]. Lastly, improving overall access to healthcare—through strengthening health systems (both better health system infrastructure and more accessible services such as community health workers) could yield enduring benefits, surpassing the impact of targeted funding for individual or vertical programmes[28,31]. A comprehensive approach to healthcare delivery that addresses systemic challenges in general access to care will promote sustainable improvements in health outcomes and reinforce the effectiveness of

other interventions like vaccines, seasonal malaria chemoprevention, IRS, and ITNs.

Finally, although urban populations benefit from greater physical access and proximity to public health services and abundant private options, the reported increase in treatment delays or lack of significant progress in reducing moderate and severe delays underscores the need for further investigation into underlying bottlenecks. Factors such as high out-of-pocket expenses (including transportation and hospital registration fees) and other barriers affecting access to health services must be addressed to improve treatment outcomes.

Beyond programmatic implications, it is also important to consider methodological aspects of our analysis in the interpretation of these findings. Potential issues of survey sample size, weighting, and representativeness were addressed by using a binomial model to estimate national-level proportions of delayed treatment, with each survey weighted according to the precision of its national estimate. This ensures that larger sample sizes from more populous countries, such as Nigeria, did not disproportionately influence the results. Each country, therefore, contributes equally in terms of national representation, regardless of sample size. The DHS and MICS surveys utilised in this study follow established sampling methodologies designed to ensure national representativeness. While survey sizes vary across countries—reflecting population differences and survey design needs—this variation does not compromise comparability. Instead, it reinforces the robustness of our results and supports their generalisability across diverse settings. This approach enhances comparability across countries and strengthens confidence that the observed patterns reflect true regional trends rather than artefacts of survey design.

### Next steps of further research

As child mortality from malaria declines in Africa, preventable deaths in children under 5 years will become increasingly concentrated in subnational areas and hotspots. As such, future research on treatment delays would benefit from increased granularity to better inform targeted local policy responses. Such research would also benefit from delving deeper into the reasons for the treatment delay, including social and cultural influences on delaying care seeking. It remains pertinent to explore further opportunities for strengthening malaria care delivery systems and the need to highlight the local bottlenecks of scaling up interventions towards improving prompt access to antimalarials, which will be critical for achieving broad child survival.

### Strengths and limitations

This study presents the most comprehensive estimates of delayed treatment with antimalarials in malaria-endemic countries in Africa. Nonetheless, the study has several limitations. First, despite the large number of surveys ($n = 177$) utilised for this analysis, responses within the survey to questions on timing to treatment were usable for only a subset (77.7%) of children captured in the surveys (111,325 out of 143,333 children). Secondly, this study was conducted at the national level, meaning we did not attempt to resolve subnational heterogeneity. Thirdly, utilising both MICS and DHS, while essential to maximise data coverage, nonetheless may have also introduced potential inconsistencies between results from the two survey sources, given subtle differences in question design and survey implementation. However, care was taken to identify and align relevant survey questions and responses to ensure that the response dataset was as consistent as possible. Fourth, although the surveys restrict the responses to only symptoms and treatments that occurred within the last two weeks, it is challenging to ascertain the accuracy of the responses provided in the data. This is because recall bias presents an inherent limitation based on the mother's or caregiver's responses to questions concerning the estimated time to receive antimalarials[32]. Furthermore, the temporal signal in the response data may also have been affected

by the WHO policy shift in 2010 from treating all child fevers in highly endemic areas with an antimalarial (and does not account for over-prescriptions) to recommending parasitological confirmation of all febrile cases suspected of malaria before treatment. This policy recommendation changed the denominator from all febrile children under age 5 to only those with confirmed malaria, although surveys continue to report widespread treatment in the absence of a diagnostic test for malaria due to RDT stockouts. Lastly, while we acknowledge the potential impact of the SARS-CoV-2 pandemic on care-seeking behaviour during the pandemic peak[33,34], long survey intervals made it difficult to distinguish between pandemic-related disruptions impact and other underlying trends, and thus beyond the scope of this study.

Despite the limitations presented above, this study provides the best available evidence on the spatial and temporal distribution of malaria treatment delay on the continent of Africa between 2006 and 2022. This study provides the most consistent and readily available treatment delay estimates and can potentially contribute to an improved understanding of severe malaria and mortality patterns while also stratifying this distribution across rural-urban and socio-economic strata. We also highlight that consideration of care-seeking as a binary event is not sufficient on its own; the timing of care is critical and can have significant implications for effective treatment.

Our study shows that, while significant progress in enhancing the overall availability and access to antimalarials among the rural and poor populations of Africa has been made, less than half of the treatment occurs <24 h of symptoms onset and 34 million malaria cases were treated only after a clinically significant delay in SSA in 2022. We have shown that the spatial patterns, temporal trends, and levels of malaria treatment delay vary between countries according to the location of residence (rural vs urban) and wealth status. This information supports drawing lessons from areas that have improved while highlighting areas needing greater attention. While the sustained progress towards eliminating and eradicating malaria may lie in discovering novel systems or technologies to build on current gains, our study illustrates the potential for optimising existing tools to help accelerate the progress against the disease. We advocate for the need to prioritise the remaining 30% of febrile children who experience severe delays in receiving antimalarial treatment. This work also illustrates that quantification of treatment rates alone, without considering the promptness of that treatment, may be insufficient for understanding the drivers of severe disease and death from malaria in children. Enacting and emphasising policies that encourage prompt treatment <24 h of symptoms must be a priority to reduce malaria mortality further and ensure accelerated progress towards international goals that target better child health.

## Methods

### Data description and sources

For consistency in terminology, this study categorises a child as receiving 'prompt' treatment if they received an antimalarial <24 h of fever onset, and as receiving 'delayed' treatment if over 24 h. We further stratified into 'moderate delay' (24 to <48 h) and 'severe delay' (>48 h). We used 177 publicly available nationally representative cross-sectional household surveys that were conducted in endemic African countries between 2006 and 2022. Each survey asked caregivers for a two-week history of fever among children under 5 years old, including the number of days from the onset of fever until antimalarial treatment was obtained. Two publicly available data sources were utilised: the Demographic Health Surveys Program[35] ($n = 117$) and the UNICEF Multiple Indicator Cluster Survey (MICS, $n = 60$). Data were collated from three DHS Programme household survey types: Malaria Indicator Surveys (MIS), Demographic and Health Surveys (DHS), and Service Provision Assessments (SPA). Further details on these data sources are provided in Supplementary Information (SI) Figs. S1, S2 and Table S1.

The combined dataset included 144,292 febrile children under 5 who received antimalarials from any source (i.e., public or private health facilities, community health workers, chemists, drug stores/sellers or pharmacies). The authors have permission for use of the data for this research purposes.

## Data processing

Survey data were weighted to derive nationally representative estimates of delayed treatment rates and corresponding sample sizes. A total of 31,806 children's observations were excluded from the analysis because they had an invalid (e.g timing response >14 days or negative values) or missing response to the question about the timing of receiving the antimalarial since the onset of symptoms. This also included survey outliers, those with inconsistent values, and those with fewer than ten weighted sample observations (SI Table S2). The inclusion criteria for the surveys are shown in SI Fig. S3. Of the 46 countries addressed in this analysis, 40 had at least one survey, and 35 had at least two surveys. The 40 countries with relevant national surveys represented 87% of the total, with the remaining six countries estimated entirely via our predictive model.

## Covariates

Candidate model covariates were selected to characterise demographic, socioeconomic, and biophysical parameters hypothesised to influence treatment delay. Relevant national-level covariates from libraries of the Institute for Health Metrics and Evaluation (IHME), the World Health Organization (WHO), and the Malaria Atlas Project (MAP)[36,37] were explored. A set of 29 covariates was initially selected and subsequently reduced by a generalised linear model via a penalised maximum likelihood-based variable selection process to yield a set of 11 predictor variables used in the final models[38]. Covariates were standardised, and outliers were removed prior to modelling. SI Table S3 shows all covariates considered for the analysis, as well as those used in the final model.

## Model fitting, validation and performance

Our model choice was based on a comparison of multiple model performance metrics, including the Watanabe-Akaike Information Criterion (WAIC), Deviance Information Criterion, Conditional Predictive Ordinate (CPO), and Marginal Likelihood (Mlik). Details of these metrics are reported in the SI, and specific results in SI Table S4. Based on the choice of the best-performing model, we modelled the fraction of delayed treatment for each country-year between 2006 and 2022 using a generalised linear mixed model implemented in a hierarchical Bayesian framework with spatio-temporal random effects. The primary response variable was the fraction of treated children for whom treatment was delayed in country $k$ at year $j$, denoted as $Y_{kj}$. This response variable was assumed to follow a binomial distribution:

$$Y_{kj} \sim \text{Binomial}\left(n_{kj}, \theta_{kj}\right) \qquad (1)$$

where $n_{kj}$ was the total number of surveyed children treated in country $k$ at year $j$, and $\theta_{kj}$ represented the underlying probability of a febrile child delaying or severely delaying receiving antimalarial treatment (i.e., >24 h or >48 h) since symptom onset within the given population. Composition of the delay and severe delay predictions enabled the derivation of the fraction of moderate delay (24 to <48 h).

To account for temporal and spatial variability and improve the robustness of our predictions, we incorporated a hierarchical structure into our model. This hierarchical structure allowed us to borrow strength across different countries and years, improving the stability and accuracy of our estimates. Specifically, we modelled the logit-transformed probabilities $\theta_{kj}$ as a function of country-specific $\alpha_k$ and year-specific $\gamma_j$ effects, as well as potential covariates $X_{kj}$ and an independent and identically distributed (i.i.d) Gaussian noise term $\epsilon_{kj}$:

$$\text{logit}\left(\theta_{kj}\right) = \alpha_k + \gamma_j + X_{kj}\beta + \epsilon_{kj}. \qquad (2)$$

$X_{kj}$ denotes a vector of covariates (e.g., socio-economic factors, health system indicators) that may influence treatment delays.

$\beta$ is the vector of coefficients associated with the covariates.

$\gamma_j$ represents year-specific effects to capture temporal trends.

For computational efficiency, the empirical logit transformation was applied to the response variable to yield the transformed variable $Z_{kj}$, which is the approximation to log odds:

$$Z_{kj} = \log\left\{\frac{Y_{kj} + 0.5}{n_{kj} - Y_{kj} + 0.5}\right\}. \qquad (3)$$

A weighted linear approximation was used where the weights were calculated as

$$W_{kj} = \left[\frac{1}{Y_{kj} + 0.5}\right] + \left[\frac{1}{[n_{kj} - Y_{kj} + 0.5]}\right]. \qquad (4)$$

The transformed variable $Z_{kj}$ was then modelled as

$$Z_{kj} = \exp\left(X_{kj}\beta\right) + S(x_k) + \gamma_j, \qquad (5)$$

where $S(x_k)$ represents the country-specific spatial random effects to capture unobserved heterogeneity across countries.

A Bayesian hierarchical linear Gaussian space-time model employing BYM (Besag, York and Mollie) conditional autoregressive random effects was fitted to the empirical-logit-transformed data to estimate the fraction of children that received delayed antimalarial treatment for each country and year. The spatial component through country-year-country ID as the grouping factor in this model specifies the conditional distributions for the spatial random effect parameter as

$$S_k|s_{\backslash k} \sim \mathcal{N}\left(\frac{1}{\sum_j w_{kj}}\sum_j w_{kj}S_j, \frac{\sigma_s^2}{\sum_j w_{kj}}\right) \equiv S_k|s_{\backslash k} \sim \mathcal{N}\left(\left\{\boldsymbol{D}^{-1}\boldsymbol{W}\boldsymbol{s}\right\}_k, \sigma_s^2\left\{\boldsymbol{D}^{-1}\right\}_{kk}\right),$$
$$(6)$$

where $w_{kj}$ is the element of a spatial weights' matrix $\boldsymbol{W}$ corresponding to row $k$ and column $j$ and $\boldsymbol{D}$ is a diagonal matrix with elements $diag(\sum_j w_{kj})$. The notation $\backslash k$ indicates neighbours of spatial units $k$. $\boldsymbol{W}$ determines the spatial proximity between the random effects, defined as a binary, first-order, adjacency matrix, whereby the spatial component of the model includes structured and unstructured spatial effects to capture spatial relationships across countries. We applied *loggamma* hyperpriors with specified parameters to control the precision of the unstructured and spatial components.

$$w_{kj} = \begin{cases} 1 & \text{if } area k and j are adjacent \\ 0 & \text{otherwise} \end{cases} \qquad (7)$$

To characterise temporal trends while accounting for the spatial variation between countries, the hierarchical time series between countries and WHO Subregions were included within the modelling framework to leverage shared autocorrelation based on the country's subregional grouping during the fitting process (see SI Fig. S4). A stationary autoregressive time-series of order one (i.e., AR1) at the country level with country-year-year ID as the time variable alongside a Markovian temporal structure was implemented. These settings assumed that the fraction of delayed treatment for year $j - 1$ influenced the rates of delay in year $j$ and accounted for temporal autocorrelation within each country-year combination while considering the group structure.

We fitted the model using Integrated Nested Laplace Approximation (INLA) using the R-INLA package[39]. Models with different formulations in the inclusion of latent variables and considering different choices of priors and hyperparameters were fitted to assess performance. The WAIC values were computed for initial model selection, with preference given to the model exhibiting the lowest WAIC value. To validate and calibrate the models, the conditional predictive ordinates and the probability integral transform (PIT) statistics were used[40,41].

For further model assessment, we used marginal predictive statistics and k-fold cross-validation with 10 folds of randomly selected hold-out groups. We compute the CPO value of each observation, given by

$$p(y_i|y_{-i}) = \int p(y_i|\theta)p(\theta|y_{-i})d\theta, \quad (8)$$

where $y_i$ is a single delay fraction observation, $y_{-i}$ is all observations excluding observation $i$, and $\theta$ are the parameters. A larger value of log $(CPO_i)$ indicates that the model assigns high probabilities to the observations, i.e., that the observation is highly likely given the data and model. The mean of the log of the CPO values for our model is -0.66; indicating generally good predictive performance and an in-sample MSE of 0.018. The k-fold cross-validation results indicate there is no substantial difference between in-sample and out-of-sample predictive accuracy. The average RMSE over the 10 folds was 0.138 for in-sample predictions and 0.15 for out-of-sample predictions. This consistency between in and out-of-sample prediction accuracy suggests that the model is not overfitting to the data.

To explore the patterns and trends in the fraction of delayed treatment in various sub-populations, we performed the modelling using i) all survey data in one unified model; subsets of data stratified according to ii) place of residence (i.e., rural vs urban populations), iii) wealth quintiles (i.e., high wealth vs lower wealth quintile households), and iv) across gender and v) age categories (i.e., younger <24 months vs older children). We also present pooled country estimates and year survey estimates derived from raw survey data, which are illustrated in SI Figs. S5-S6. Modelled estimates of the fraction of treatment delay for countries without any survey data were derived from the learned relationships between the fraction of delay and the covariates, leveraging spatial and temporal autocorrelation within the modelling framework.

### Presentation of the estimates

Modelled posterior summaries were obtained for the distribution's selected quantiles (0.025, 0.25, 0.5, 0.75, 0.975). We also report the full uncertainty distribution, with estimates for all metrics computed using 10,000 realisation draws from the Bayesian posterior predictive distribution and present 95% uncertainty intervals (UI) given as the 2·5 and 97·5 percentiles of that distribution. The continental, subregional, and national estimates of delayed treatment are given for the period 2006-2022, and maps indicating spatial distribution and temporal patterns are presented. Using previously published malaria case estimates[19], we estimated the number and fraction of clinical malaria cases receiving delayed treatment per country-year. All data curation, modelling, plotting and visual presentation of inputs (Figures S1–S7) and outputs (Figs. 1–4) were implemented in R version 4.21[42] and ArcGIS Pro 10.8.1. While the modelled outputs will be freely available to the public, raw data will not be shared unless with express permission from the DHS and UNICEF.

### Reporting summary

Further information on research design is available in the Nature Portfolio Reporting Summary linked to this article.

## Data availability

All processed and modelled delay estimates data generated in this study have been deposited in the Zenodo database under accession https://doi.org/10.5281/zenodo.17096726. The raw data are publicly available through the Demographic and Health Surveys (DHS) Programme (https://dhsprogram.com/) and the UNICEF Multiple Indicator Cluster Surveys (MICS) (https://mics.unicef.org/). Access to these datasets requires free registration and approval from the respective data providers. Additional data are provided in the Supplementary Information/Source Data file. All country-level model results are included in the Supplementary Information files. The geographic shapefiles used to generate mapped results are available for download through the Malaria Atlas Project R package (https://cran.r-project.org/web/packages/malariaAtlas/index.html).

## Code availability

The code for the analysis in this paper is available from https://github.com/Jailos/treatment-delay.

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

## Acknowledgements

This work was supported, in whole or in part, by the Bill & Melinda Gates Foundation [INV-055192 and INV-009390/OPP1197730]. The conclusions and opinions expressed in this work are those of the author(s) alone and shall not be attributed to the Foundation. Under the grant conditions of the Foundation, a Creative Commons Attribution 4.0 License has already been assigned to the Author Accepted Manuscript version that might arise from this submission. Please note that works submitted as a preprint have not undergone a peer review process. This work also includes funding support from the Australian Government, National Health and Medical Research Council (Award No: GNT2025280) and Telethon Trust, Western Australia. Funders of this work had no role in study design, data curation, analysis, interpretation, or the writing of this manuscript. The corresponding author had full access to all the data in the study, and all authors had the final responsibility for the decision to submit for publication.

## Author contributions

J.L. and S.F.R. conceptualisation, study design, data curation, formal analysis and wrote the manuscript; P.A.D., M.M., A.S., A.B., F.S., Y.G., P.A., T.L.S., E.C., J.B.K.; data interpretation, and provided internal review & editing; P.C., J.H., J.A.R. and C.V. extracted the data, developed and maintained the databases of response data and created the maps; D.J.W. and P.W.G.—funding acquisition, supervision, review & editing. All authors have reviewed and contributed to the manuscript.

## Competing interests

The authors declare no competing interests.
