## [Transparent Peer Review file · Nature Communications]

Estimating rates of treatment delay for malaria fevers among children in Sub-Saharan Africa 2006-2022

Corresponding Author: Dr Jailos Lubinda

Version 0:

Reviewer comments:

Reviewer #1

(Remarks to the Author)

The authors use data from national surveys to estimate the proportion of children experiencing delays in receiving antimalarial treatment across the African continent from 2006 – 2022. Delayed treatment is defined as >24 hrs from fever onset, while severe delay is defined as >48 hrs from fever onset. Trends across time are reported, as are differences between countries. Given that treatment delays are almost certainly a key contributor to morbidity and mortality, this study makes a worthwhile contribution to the literature.

Although the manuscript is generally well written, there are sections throughout the introduction/results/discussion that are quite repetitive, with several points being made repeatedly. I have listed a few (not all) examples below.

Specific comments:

Introduction:

- There are several statements throughout the introduction that lack references to support the specific claims. Eg, line 39: the statement that “treatment after a 24 hr delay... is a common trigger for malaria outbreaks in low transmission settings” requires a reference. Particularly given that the sentence begins “Evidence shows...”
- Line 39 – 40. The sentence beginning “prompt treatment within 24 hours is reported to...” also requires a reference.
- Line 43 - 45 – The reference provided (Kamabai et al) does not support the statement made in this sentence,
- Line 47 – suggest replace “drives” with “increased”
- Line 48 – “the literature”, rather than just “literature”.
- Line 51-52 – this sentence is repetitive – the link between delayed treatment and increased risk of severe malarial anaemia has already been stated in the 1st paragraph.
- Line 54-55 – the meaning of this sentence is not clear.
- Line 60 – suggest specify that this is children in Africa (because elsewhere, ie. in low transmission settings, adults have a higher risk of severe disease)
- Line 67 – please define “SSA” the first time it is used.
- Last sentence in intro – suggest reword, as currently it reads as if this is the first systematic review since 2006 (ie, implying that there may have been another survey prior to this).

Results:

- For this particular paper it would work better for the methods to be placed above the results. If journal requirements prohibit this, then the authors will need to include key methodological details within the results. Otherwise, the results cannot be adequately understood. As a minimum, the authors need to define “delayed treatment” and “severe delay”. They should also include a brief description of the source of the data, ie. what were the surveys, how many countries were surveyed, how many febrile episodes, etc.
- Line 81 – please state what the WHO recommendations are
- Line 82 – when stating the range, suggest list the smaller number first.
- Line 87 – can the authors include a P value for the fall in treatment delay from 2008 to 2022? Also, this result has already been reported in line 78 – except the UI in line 78 is 45.7 – 72.8, whereas in line 87 it is 45.7 – 71.8.
- All acronyms need to be defined the first time they are used, ie. AFRO-W, EMRO etc. Also please be consistent (eg. Line 89 “AFRO-W”; line 97 “west Africa”; line 95 “West Africa”;

- In the figures the authors use “CI (credible intervals)” while in the text they use “UI (uncertainty intervals”).
- Please be consistent with whether using words (eg “four percent” in line 95) or figures (eg 1% in line 99).
- Figure 2 – please define all acronyms in legend
- Line 108 – these data have already been presented – ie in line 87 it is stated that 60% of children experience treatment delay in 2022, while in line 108 it is stated that 41% received prompt treatment. These are the same results.
- Line 141 – what is meant by “poorer administrative units progressed”?
- Line 145 – replace “are” with “were”
- The authors discuss the different fractions of treatment delay in rural vs region, and higher vs lower wealth quintile. Was it possible to conduct any multivariate analysis to identify predictors of delayed treatment?

Discussion:

- Line 160 – I suggest the authors refer to their study as a study, rather than as an “exercise”
- Line 162 - suggest add “likely contributes to” since, the study did not actually evaluate severe malaria and malaria mortality.
- Line 181 – suggest add these data to the results text.
- Line 191 – 203 – in this paragraph the authors discuss the absolute number of malaria deaths in each country. They acknowledge that this absolute number will be influenced by the population, but can they not calculate (or at least estimate) the mortality rate, ie. per malaria case? This would then allow them to comment on any apparent correlation between treatment delays and mortality rates, which would be an important finding.
- Line 222 – please provide a reference for these “current guidelines”
- Lines 248 – 249 – this information should be included in the results text.

Reviewer #2

(Remarks to the Author)

Please see the attached report.

Reviewer #3

(Remarks to the Author)

This study, a secondary analysis of data from available from representative national surveys across Sub-Saharan Africa, describes that spatial distribution of delays in accessing healthcare for malaria from a variety of angles and across time. Overall, it was estimated that there was a relatively even split in children receiving timely care, moderately delayed care, and severe delays in care across the continent. The authors show how early progress was made in many countries to reduce delays from 2008 to 2014, but by 2015 this process had stagnated and, in some countries, actually began to revert to increasing delays. Overall, the greatest delays were seen in the Central African and Eastern Mediterranean regions, particularly Somalia, Camarron, and the DRC. This is in contrast to the East African region, where the most improvement was seen particularly in Tanzania and Burundi, in which did not see the same stall in progress.

This manuscript covers an important and sparsely addressed topic. This is an interesting study that provides a good background for more granular work as it is limited in its spatial resolution. While there is in-depth analysis and a robust supplement, there are remaining questions and clarity needed for some of the methods.

Major comments:

It is not clear to me why this analysis is not conducted at the individual level, which would reduce the risk of ecological fallacy and also allow the authors to control for known individual-level risk factors for malaria, including severe malaria. For example, the DHS surveys include data that is more directly related to malaria treatment access than the suite of IHME covariates, such as the source of antimalarial treatment women have during pregnancy, which would arguably be a better indicator of access to antimalarials than the national-level covariates. There are a number of covariates related to accessing health facilities in the DHS survey, which would be useful in understanding access to care and/or delayed antimalarial treatment. My recommendation is that the authors revisit the analysis and consider constructing a DAG about the risk factors that may be driving delayed treatment, rather than rely on variable-selection of a suite of covariates that are available from IHME/MAP/etc and are not necessarily aligned with the survey data or the research question. To illustrate my point, consider the following: if the proportion of skilled birth attendants is a selected and important covariate that is relevant for malaria treatment delays in the model, then wouldn't a mother's access to antimalarials during pregnancy be a more important covariate, since it more directly measures access to antimalarials? A mother's access to antimalarials during pregnancy is available from DHS surveys, and the authors could even estimate the national proportion, which would be consistent with the analysis they've done.

More generally, I don't see the utility of using an approximation to the log odds via the variable Z. The number of countries and time periods and observations in the data does not warrant this approximation, since this model should be able to be fit in INLA on a standard laptop. Can the authors refit to a binomial outcome?

Line 90: Given there are only two EMRO countries with malaria in this analysis and that these countries are geographically mixed with East Africa, is it included as a separate region for the main purpose of being consistent with the WHO?

Line 110: When you talk about the percentage of children who received an antimalarial falling into different treatment categories, why use receipt of antimalarials in particular? Is this just a way to say children with malaria that were treated or is receipt of antimalarial used as a proxy for having malaria? If so, how are you accounting for over-prescription of antimalarials?

Lines 223-229: The authors make a number of recommendations to reduce delayed treatment. However, they provide no citations to suggest that these recommendations will lead to the best health outcomes. Are these meant to be hypotheses about potential solutions to the problem, or is there a literature to draw upon when making these recommendations.

Line 305: What would be considered an invalid response to the question about timing or inconsistent values?

Line 314: Why were outliers removed when the aim of the paper was to characterize delays in care seeking across Africa? Was there an investigation into why a country or row of data might be an outlier? Or are the authors using a pure statistical approach to "outlier" detection. If the latter, this may lead to smoother model estimates, but not necessarily more robust and/or better evidence. After all, it's entirely possible that those 'outliers' are those most in need of reliable and timely access to antimalarials.

Line 324: Was this the process for estimating severe delays as well? How were the survey questions used to calculate these delays formatted/what did they explicitly ask?

Line 348-49: What is country-year-country ID? The way the model is written in line 343 implies independent spatial and temporal processes. The former is implemented through a CAR prior, while the latter an AR1 process. The two independent processes imply that the spatial process is constant over time, and the temporal process is constant over space. But the labeling here seems to imply a separable (or non-separable) process, which is confusing to me. Can the authors clarify this?

Line 362: What is country-year-year ID? Same as above.

(Supplemental Material) Page 11 Figure S3: Were children who were never treated considered? While they would not have data on time to antimalarial, those children would still not be receiving timely care.

Line 90: Given there are only two EMRO countries with malaria in this analysis and that these countries are geographically mixed with East Africa, is it included as a separate region for the main purpose of being consistent with the WHO?

Minor Comments:

Figure 3: It would be useful if the number of cases treated that were not severely or moderately delayed were added to this figure. Also, the caption states the x-axis includes number of children with antimalarials, but I believe the authors mean the y-axis.

Line 59: The reported number of deaths should be updated to the most recent world malaria report (as opposed to the 2022 report).

Lines 39-40: the statement that prompt treatment within 24 hours could potentially avert half of severe cases needs a citation.

Line 48: 'hours' instead of 'hrs'

Line 108: 'approximately over a third' is awkward. Why not just state 'an estimated 41%'...

Reviewer #4

(Remarks to the Author)

Version 1:

Reviewer comments:

Reviewer #2

(Remarks to the Author)

The authors have addressed my concerns. However, the Discussion section of the manuscript requires substantially more text added to the Discussion section to reflect the responses to comments 2.1 and 2.2 from Reviewer 2 so that future readers of the article will fully understand the methods employed.

Reviewer #3

(Remarks to the Author)

We thank all the four reviewers for their thorough review and constructive feedback of our manuscript content. Below, we detail our responses to each point raised. Reviewer comments are given in black, with our responses in orange.

Responses to Reviewer #1

1. **Comment:** There are several statements throughout the introduction that lack references to support the specific claims. Eg, line 39: the statement that “treatment after a 24 hr delay... is a common trigger for malaria outbreaks in low transmission settings” requires a reference. Particularly given that the sentence begins “Evidence shows...”
 - **Response:** We now cite the following study:
Freeman, J., Laserson, K. F., Petralanda, I., & Spielman, A. (1999). Effect of chemotherapy on malaria transmission among Yanomami Amerindians: simulated consequences of placebo treatment. The American journal of tropical medicine and hygiene, 60(5), 774-780.
2. **Comment:** 2. - Line 39 – 40. The sentence beginning “prompt treatment within 24 hours is reported to...” also requires a reference.
 - **Response:** We have added the citation supporting the importance of prompt treatment. We now cite the following study:
Mousa, A., et al. The impact of delayed treatment of uncomplicated P. falciparum malaria on progression to severe malaria: A systematic review and a pooled multicentre individual-patient meta-analysis. PLoS medicine 17, e1003359-e1003359 (2020).
3. **Comment:** Line 43 - 45 – The reference provided (Kamabai et al) does not support the statement made in this sentence
 - **Response:** Thank you for spotting this referencing error. The sentence was supposed to be attributed to the previously used reference (White N, 2022), and we have now corrected the citation.
4. **Comment:** Line 47 – suggest replace “drives” with “increased”
 - **Response:** We agree with your suggestion. However, this has been superseded by the revisions of the sentence at another reviewer's suggestion.
5. **Comment:** Line 48 – “the literature”, rather than just “literature”.
 - **Response:** This has been revised now
6. **Comment:** Line 51-52 – this sentence is repetitive – the link between delayed treatment and increased risk of severe malarial anaemia has already been stated in the 1st paragraph.

- **Response:** We have revised the paragraph and removed any obvious repetitions to avoid redundancy and streamline the link between delayed treatment and severe malaria outcomes.
7. **Comment:** Line 54-55 – the meaning of this sentence is not clear.
- **Response:** We have revised this sentence for better clarity and flow, and it now reads:

"Recent multisite studies on malaria mortality across Kenya, Mali, Mozambique, and Sierra Leone show how delayed treatment remains highly prevalent and consequential, reporting that over 62% of malaria deaths in under-5 children had received an antimalarial before death, but too late for it to be effective"
8. **Comment:** Line 60 – suggest specify that this is children in Africa (because elsewhere, i.e. in low transmission settings, adults have a higher risk of severe disease).
- **Response:** As suggested, we have specified that the data pertains to children in Africa to distinguish the age-related risks of malaria.
9. **Comment:** Line 67 – please define “SSA” the first time it is used.
- **Response:** We have now defined "SSA" as "Sub-Saharan Africa" upon its first use.
10. **Comment:** Last sentence in intro – suggest reword, as currently it reads as if this is the first systematic review since 2006 (ie, implying that there may have been another survey prior to this).
- **Response:** We appreciate the reviewer’s observation and now add more specific detail to this paragraph to avoid any ambiguity. We also continue to avoid the use of the term ‘systematic review’.
10. **Comment:** 1. - For this particular paper it would work better for the methods to be placed above the results. If journal requirements prohibit this, then the authors will need to include key methodological details within the results. Otherwise, the results cannot be adequately understood. As a minimum, the authors need to define “delayed treatment” and “severe delay”. They should also include a brief description of the source of the data, ie. what were the surveys, how many countries were surveyed, how many febrile episodes, etc.
- **Response:** We appreciate the feedback and confirm that we have structured the paper according to *Nature Communications* guidelines. We have followed the reviewer’s suggestion of augmenting the Introduction section with some additional key details on the methodological approach, to facilitate the reader’s interpretation of the Results that directly follow.
11. **Comment:** Line 81 – please state what the WHO recommendations are

- **Response:** Thank you for the observation. Although this was defined/shown in the original lines 221, we have now explicitly stated the WHO recommendation for prompt treatment of malaria.

12. **Comment:** Line 82 – when stating the range, suggest list the smaller number first.

- **Response:** We have revised the sentence to list the range in the correct order, with the smaller number first, for consistency and clarity.

13. **Comment:** Line 87 – can the authors include a P value for the fall in treatment delay from 2008 to 2022? Also, this result has already been reported in line 78 – except the UI in line 78 is 45.7 – 72.8, whereas in line 87 it is 45.7 – 71.8.

- **Response:** We thank the reviewer for spotting the typo in the UI. We have corrected the discrepancy in the uncertainty intervals in lines 78 and 87 to ensure consistency.
- Regarding the *p-value*: Since our analysis uses Bayesian inference, frequentist p-values are not applicable. Instead, and in line with the convention for this type of analysis, we report the posterior probability that treatment delay declined from 2008 to 2022, along with the 95% uncertainty interval (UI) for the estimated change.

14. **Comment:** All acronyms need to be defined the first time they are used, ie. AFRO-W, EMRO etc. Also please be consistent (eg. Line 89 “AFRO-W”; line 97 “west Africa”; line 95 “West Africa”;

- **Response:** We have defined all acronyms at first time they appear in the manuscript and have ensured consistent usage throughout the text.

15. **Comment:** In the figures the authors use “CI (credible intervals)” while in the text they use “UI (uncertainty intervals”).

- **Response:** We appreciate this observation. We have standardised the terminology used to “uncertainty intervals (UI)” throughout the manuscript and Supplementary Figures.

16. **Comment:** Please be consistent with whether using words (eg “four percent” in line 95) or figures (eg 1% in line 99).

- **Response:** Thank you for the observation, we have now ensured consistency in reporting percentages using figures and have made the revision throughout the manuscript accordingly.

17. **Comment:** Figure 2 – please define all acronyms in legend

- **Response:** We have updated the figure legend to include definitions for all acronyms in Fig 2

18. **Comment:** Line 108 – these data have already been presented – ie in line 87 it is stated that 60% of children experience treatment delay in 2022, while in line 108 it is stated that 41% received prompt treatment.

- **Response:** We appreciate the reviewer’s observation. The repeated data point was intentionally included to provide continuity and context for introducing the three-panel discussion, which categorizes treatment into three groups: prompt treatment (previously defined) and two subcategories of delayed treatment. Reiterating this information ensures clarity and avoids introducing new concepts without linking them to the previously presented context. To maintain logical flow and prevent potential confusion for the reader, we have opted to retain the current phrasing.

19. **Comment:** Line 141 – what is meant by “poorer administrative units progressed”?

- **Response:** Thank you for pointing this out. The phrase "poorer administrative units progressed" refers to improvements in how promptly children from low socioeconomic status (SES) areas/families received treatment for malaria. To enhance clarity, we have revised the sentence to explicitly state, “...children from poorer households or administrative units progressed in...” to ensure the meaning is clear.

20. **Comment:** Line 145 – replace “are” with “were”

- **Response:** We have corrected the verb tense to "were" as suggested.

21. **Comment:** "12. - The authors discuss the different fractions of treatment delay in rural vs region, and higher vs lower wealth quintile. Was it possible to conduct any multivariate analysis to identify predictors of delayed treatment?"

- **Response:** We appreciate this comment. The primary objective of this study was to build a predictive framework for accurate reconstruction of delay rates across geography and through time and not to offer insight into the causal factors. However, we did perform a simple multivariate analysis using a generalised linear mixed model (GLM) as part of the exploratory data analysis. The analysis, based on pooled data, identified residence, wealth, mother’s age, and education as key predictors of treatment delay. Urban residence, higher wealth, and education were associated with reduced treatment delay, while older age, multiple children under 5, and living in West and Central Africa were linked to increased delays. The child's sex and prior treatment status did not show significant effects and there was notable regional variation across countries. We used this exploratory analysis to identify plausible covariates for consideration in the final predictive modelling at the aggregate level using the IHME and MAP datasets. We have added further details on this exploratory multivariate analysis to the supplementary information.

22. **Comment:** Line 160 – I suggest the authors refer to their study as a study, rather than as an “exercise”

- **Response:** Thank you, we have adopted this phrasing.
23. **Comment:** Line 162 - suggest add “likely contributes to” since, the study did not actually evaluate severe malaria and malaria mortality.
- **Response:** We have revised the sentence to include "likely contributes to" according to the suggestion.
24. **Comment:** Line 181 – suggest add these data to the results text.
- **Response:** Thank you for the suggestion. We have moved the relevant information from this section to the results section to provide a more comprehensive presentation of our findings. The remaining sentence has been revised to ensure clarity and coherence.
25. **Comment:** Line 191 – 203 – in this paragraph the authors discuss the absolute number of malaria deaths in each country. They acknowledge that this absolute number will be influenced by the population, but can they not calculate (or at least estimate) the mortality rate, ie. per malaria case? This would then allow them to comment on any apparent correlation between treatment delays and mortality rates, which would be an important finding.
- **Response:** We appreciate the reviewer’s suggestion. We considered exploring mortality rates, but a few challenges we faced included the many versions/sources of these. While we found a strong correlation between treatment delays and mortality rates during our analysis, we deliberately chose to exclude this from the manuscript to maintain focus on the primary message of space-time patterns of delay. Additionally, as this is a national-level study, correlating delays with mortality rates may be less meaningful without subnational granularity. We believe this correlation would be more appropriately addressed in a follow-up study focusing on subnational scope.
26. **Comment:** Line 222 – please provide a reference for these “current guidelines”
- **Response:** We have included a reference for current guidelines on malaria treatment:
WHO. (2015). Guidelines for the treatment of malaria, 3rd edition. World Health Organization.
27. **Comment:** Lines 248 – 249 – this information should be included in the results text.
- **Response:** We have moved the relevant information into the results section as suggested: “Of the 46 countries addressed in this analysis, 40 had at least one survey, and 35 had at least two surveys. The 40 countries with relevant national surveys represented 87% of the total, with the remaining six countries estimated entirely via our model.”

Responses to Reviewer #2

Lubinda *et al.* have conducted an analysis of treatment delay for malaria fevers among children in sub-Saharan Africa that uses data collected from DHS and MICS surveys between 2006 and 2022. The authors should be commended for highlighting the dearth of longitudinal data available for the treatment delay related to malaria fevers among children in sub-Saharan Africa. However, the authors demonstrate the shortcomings in data by attempting this analysis with insufficient data. Consider the following:

The authors report having drawn data from 183 surveys. This appears to be incorrect since supplementary file lists each DHS and MICS survey included and excluded. The total number of surveys identified was 183 and 6 surveys were excluded (Table S2), leaving 177 surveys for analysis. Of the 177 were included, 30 had fewer than 100 children (weighed). More than half, 55.8%, of the children used for this analysis were from 16 countries where 38 surveys had weighted sample sizes that ranged from 1,005 to 3,703.

Focusing on just this majority of surveys, 38 from 16 countries, for the 2006 to 2022 time period –

- **8 countries had 1 survey:**

Benin, Burkina Faso, Burundi, Kenya, South Sudan, Tanzania, Mozambique, and Zambia;

- **2 countries had 2 surveys:**

Central African Republic, and Liberia;

- **3 countries had 3 surveys:**

Chad, Democratic Republic of the Congo, and Sierra Leone;

- **2 countries had 5 surveys:**

Malawi and Uganda;

- **1 country with 7 surveys:**

Nigeria.

Malawi, Uganda, and Nigeria contributed a majority of children, 56.1%, in surveys that were in the surveys that had sample sizes ranging from 1,005 to 3,703. Even though sample sizes were weighted according to their sample size, these 3 countries contributed nearly onethird, 31.3%, of all children for the entire analysis. This is highly problematic for an analysis that reports to generate 17 annual estimates of treatment delay in 3 intervals, i.e. <24 hours, 24 to <48 hours, and >48 hours, across 46 countries. The authors should narrow the scope of their analysis by reducing the number of countries and use this as a call to action.

General Response: We thank the reviewer for their careful reading and thorough evaluation of our manuscript. Below are our point-by-point response towards our addressing these comments.

1. **Comment:** The authors report having drawn data from 183 surveys. This appears to be incorrect since supplementary file lists each DHS and MICS survey included and excluded. The total number of surveys identified was 183 and 6 surveys were excluded (Table S2), leaving 177 surveys for analysis.

- **Response:** Thank you for pointing out this discrepancy. We confirm that the correct final number of surveys included in the models is 177, after excluding the 6 surveys mentioned in Table S2. We have updated the manuscript to clarify.

2. **Main comments** regarding data volume and representativeness:

2.1: Comment on the survey sample sizes, data weighting and concerns on survey representativeness:

- **Response:** Our analysis employs a binomial model where we estimate national proportions of delayed treatment. Nationally representative surveys are each given broadly equal weighting (influenced only by the precision of the reported national level figure) – not, as the reviewer suggests, a weight that is proportional to the number of individuals involved in each survey. As a result, the sample sizes from countries with larger populations (e.g., Nigeria) do not inherently lead to bias in our results. Consequently, the concerns about overrepresentation due to large sample sizes in certain countries, such as Nigeria, do not impact the results, as smaller countries are equally represented in terms of the proportion of delayed treatment.

2.2 Comment on the role of country sample sizes in surveys:

- **Response:** We also wish to clarify that the sampling methodology of the DHS and MICS surveys follows a well-established approach that ensures the representativeness of the populations surveyed. The larger sample sizes from more populous countries like Nigeria reflect the need to ensure a representative sample within those countries, which is not problematic in this context. Smaller countries, with smaller populations, naturally contribute fewer children to their survey, and the sample sizes are sometimes driven by the power calculations of other additional indicators collected, this variation is inherent in the design of these large-scale national surveys and doesn't affect this study, but they add confidence for generalisation.

2.3 Comment On narrowing the scope of the analysis:

- **Response:** We respectfully disagree with the recommendation to narrow the scope of our analysis based on these sampling concerns. The inclusion of a broad range of countries, including those with varying sample sizes, is crucial to capture a comprehensive picture of treatment delays in malaria care across Sub-Saharan Africa. Rather than restricting the analysis, we believe the findings highlight the importance of addressing gaps in the data. We acknowledge that there is a call for more longitudinal data, particularly for countries with fewer surveys, but reducing the number of countries would undermine the goal of obtaining

regional estimates and sustain the persisting lack of information on the subject in countries with fewer or no surveys at all.

We therefore stand by the methodology used and believe that the proportions-based analysis, along with the robust sampling approach of the DHS and MICS surveys, appropriately captures treatment delay across the region. We, however, clarify the methodology in the manuscript to ensure these points are communicated more clearly to the reader and hope our response helps clarify our approach and addresses the reviewer's concerns.

3. Other Comments:

3.1. Comment: Use consistent time periods rather than switching between 'same day' and '<24 hours'; 'next day' and '24 to <48 hours', etc. If the survey questionnaires are different, consider the category that is inclusive of either/or, i.e. 'same day / <24 hours'; 'next day / 24 to <48 hours', etc.

- **Response:** We thank the reviewer for this valuable suggestion regarding the use of consistent time-periods. We have revised the manuscript to standardize the time categories throughout. Specifically, we ensured that all terms are consistent
 - "<24 hours"
 - "24 to <48 hours"
 - ">48 hours"

By adopting this consistent terminology, we ensure that the categories are clear and inclusive, regardless of slight variations in how different surveys phrased the time frames. We have made these revisions throughout the manuscript and updated any tables and figures accordingly.

3.2. Comment: Figure 1a - The key has categories that are not particularly useful and lack in precision. Consider making cut-offs covering 5% points.

- **Response:** We thank the reviewer for their insightful comment regarding the categories in the key for Figure 1a. To improve clarity and utility, we have revised and updated the figure by adjusting the cut-offs to cover 5 percentage points as suggested. We also provide tabular data by country, in Table 1.

3.2. Comment: Figure 1a - b. Flip the order of this so 'within 24 hours' is at the top and '>48 hours' is at the bottom.

- **Response:** Thank you for your suggestion. We have now re-ordered the legend as suggested.

3.3. Comment: How did the SARS-CoV-2 pandemic affect care-seeking? This merits discussion.

- **Response:** We appreciate the reviewer’s suggestion to discuss the impact of the SARS-CoV-2 pandemic on care-seeking behaviour. The pandemic caused significant disruptions to healthcare services, mobility, and health-seeking behaviours, which may have influenced the trends observed in our post-pandemic data.

Urban areas were more affected by health system disruptions due to lockdowns, potentially exacerbating delays in care. Evidence shows that COVID-19 led to postponed or cancelled malaria intervention campaigns and delayed treatment (Rogerson et al., 2020; Dzianach et al., 2023). However, assessing its direct impact on treatment delays in our study is challenging, as many surveys were postponed or cancelled during the pandemic, and health survey data collection typically focuses on treatment within the past 14 days. Given the long survey intervals, distinguishing between pandemic-related disruptions and other underlying trends is difficult. Thus, while we acknowledge its potential impact, disentangling these effects is beyond the scope of this study.

We have added a sentence in the limitations section addressing the potential influence of the COVID-19 pandemic on observed trends, citing relevant studies on healthcare disruptions caused by pandemic-related restrictions, fear of infection, and system strain. The sentence reads as follows:

“Finally, due to the small number of surveys after 2019, the estimates for 2020 treatment-seeking rates are likely to not reflect the disruptions to healthcare utilisation caused by the COVID-19 pandemic (Rogerson et al., 2020; Dzianach et al., 2023).”

3.4. Comment: The authors should explore the drivers of care-seeking in East Africa among households of low SES and rural areas. The downward trend is unlike any other in any sub-region.

- **Response:** We thank the reviewer for suggesting a deeper exploration of the drivers of care-seeking in East Africa, particularly in households of low socioeconomic status (SES) and rural areas. We have expanded the discussion in the manuscript to explore potential drivers that may explain the observed patterns. We incorporate relevant studies that provide a more comprehensive understanding of why the trend in East Africa diverges from those observed in other sub-regions. The incorporated text reads as below:

The faster decline of delayed treatment in rural and low SES households of East Africa compared to urban areas or high SES may reflect a prevailing complex interplay of healthcare access and socio-demographic factors. Expanding urban slums in East Africa face worsening health conditions due to rapid urbanisation, overcrowding, and inadequate healthcare, leading to a slower decline in health outcomes and reversing the "urban advantage" traditionally associated with better access to services (Kimani-Murage et al., 2014; Matthews et al., 2010). In contrast, rural areas have significantly benefited from targeted health interventions, community-based care, and improved outreach, resulting in

better care-seeking behaviour and reduced delays (Lungu et al., 2018; Macharia et al., 2023). In countries such as Kenya, Tanzania, Ethiopia, and Uganda, neonatal mortality rates have also declined faster in rural settings, narrowing the urban-rural gap and highlighting poor infrastructure and limited access to care as key drivers of slower progress in urban areas (Matthews et al., 2010, Kimani-Murage et al., 2014, Macharia et al., 2023, Norris et al., 2022).

3.4. Comment: Abstract: Lines 22-24: The estimates of delay need to include the credible intervals included for 60%, 29%, 76%, and 35.3%

- **Response:** We appreciate the reviewer's suggestion to include credible intervals for the estimates of delay. We have updated the abstract to include the credible intervals for the estimates of treatment delay, specifically for the figures mentioned (60%, 29%, 76%, and 35.3%).

3.4. Comment: Introduction Line 33-34: The statement, "Since the African Summit on Roll Back Malaria's Abuja Declaration, it has been established that early diagnosis and treatment within 24 hours of the onset of symptoms is the optimal strategy for uncomplicated malaria." Is not really correct. This has always been optimal. The RBM Abuja Declaration focused the international community around the importance of early diagnosis and treatment. The statement should be rephrased.

- **Response:** We have revised this sentence to more accurately reflect that the Abuja Declaration helped to focus global attention on the critical need for early diagnosis and treatment within 24 hours of symptoms, which has always been the optimal strategy for malaria management. The revised sentence reads:

"Since the African Summit on Roll Back Malaria's Abuja Declaration, there has been a renewed global emphasis on the importance of early diagnosis and treatment <24 hours of the onset of symptoms, which is the optimal strategy for managing uncomplicated malaria."

3.5. Comment: Line 37-38: Provide the reported range of odds for severe malaria attributable to a 2- 3 day delay in care-seeking rather than stating, "...up to 2.8 times..."

- **Response:** We have revised the sentence to include the reported specific range of odds for severe malaria attributable to a 2-3 day delay and more, based on the relevant literature. The revised sentence now reads: "...is associated with odds ratios of 2.79 times for severe malarial anaemia in children delayed by 2–3 days, 5.46 for a delay of >7 days"."

3.6. Comment: Line 38: Define 'low transmission settings'. Please define 'low' within the sentence.

- **Response:** We have defined "low transmission settings" in the manuscript to ensure readers understand the context. The revised sentence reads:

"...low-transmission settings (<10% PfPR₂₋₁₀)". This is the definition according to the WHO criteria.

3.7. Comment: Line 47-48: The statement, "and potentially drives drug resistance due to delayed parasite clearance." should elaborate. Delayed care-seeking will lead to an increase in parasite carriage which will, in turn, necessitate more treatment days to clear infection.

- **Response:** We appreciate the reviewer's suggestion to elaborate on the link between delayed care-seeking and drug resistance. We have expanded on this point to clarify that delayed care-seeking leads to prolonged parasite carriage, which in turn increases the likelihood of suboptimal drug responses and treatment failure, contributing to the development of resistance. The revised sentence reads:

"Other potential consequences include increases in malaria reservoirs for onward transmission to mosquitoes, leading to prolonged parasite carriage, which increases the risk of drug resistance due to suboptimal parasite clearance and may compromise antimalarial drug effectiveness in routine healthcare settings."

Responses to Reviewer #3

1. **Comment:** It is not clear to me why this analysis is not conducted at the individual level, which would reduce the risk of ecological fallacy and also allow the authors to control for known individual-level risk factors for malaria, including severe malaria. For example, the DHS surveys include data that is more directly related to malaria treatment access than the suite of IHME covariates, such as the source of antimalarial treatment women have during pregnancy, which would arguably be a better indicator of access to antimalarials than the national-level covariates. There are several covariates related to accessing health facilities in the DHS survey, which would be useful in understanding access to care and/or delayed antimalarial treatment. My recommendation is that the authors revisit the analysis and consider constructing a DAG about the risk factors that may be driving delayed treatment, rather than rely on variable selection of a suite of covariates that are available from IHME/MAP/etc and are not necessarily aligned with the survey data or the research question. To illustrate my point, consider the following: if the proportion of skilled birth attendants is a selected and important covariate that is relevant for malaria treatment delays in the model, then wouldn't a mother's access to antimalarials during pregnancy be a more important covariate since it more directly measures access to antimalarials? A mother's access to antimalarials during pregnancy is available from DHS surveys, and the authors could even estimate the national proportion, which would be consistent with the analysis they've done.

- **Response:** We appreciate the reviewer's comment and suggestion to conduct an individual-level analysis. However: the crucial distinction here is that the purpose of our analysis is *predictive* rather than *inferential*. In other words, the purpose of this work is to generate robust estimates of delay rates – geographically and temporally – in order to provide policy makers with improved situational awareness of this urgent public health challenge. The work is not motivated by a desire to better understand which causal factors determine delayed treatment, nor quantify their respective importance. The suggestion of conducting an individual-level analysis is exactly correct if our aim was the latter but would simply be unsuitable for the former. Use of the individual-level survey data for covariate information, rich though it is, provides no means of then generating predictions for country-years without surveys (the vast majority). This is why we instead use the aggregate-level covariates from our own MAP and IHME libraries, which are available for all geography-years.

2. **Comment:** More generally, I don't see the utility of using an approximation to the log odds via the variable Z. The number of countries and time periods and observations in the data does not warrant this approximation, since this model should be able to be fit in INLA on a standard laptop. Can the authors refit to a binomial outcome?

- **Response:** We appreciate the reviewer's concern regarding the use of an approximation to the log odds via the variable Z. To address this, we explored multiple alternative models, including direct binomial models. Our model selection was based on multiple performance metrics, including Watanabe-Akaike Information Criterion (WAIC), Deviance Information Criterion (DIC), Conditional Predictive Ordinate (CPO), and marginal likelihood (Mlik). The results, summarized in

SI Table S4, indicate that the current log odds model consistently outperformed the alternatives across these criteria, with the exception of the Probability Integral Transform (PIT), where the models exhibited very similar fits. Given these findings, we maintain the selected model and have updated the manuscript accordingly to reflect the model choice based on the reported model performance comparison.

3. **Comment:** Line 90: Given there are only two EMRO countries with malaria in this analysis and that these countries are geographically mixed with East Africa, is it included as a separate region for the main purpose of being consistent with the WHO?
 - **Response:** Thank you for your comment. As noted, there are three EMRO countries with malaria (Djibouti, Sudan, and Somalia) in this analysis. We chose to follow the WHO regional classifications to maintain consistency with reporting frameworks. While we acknowledge the geographic overlap with East Africa, this approach allows for better comparability with other WHO-based analyses and ensures alignment with standard regional categorisation practices.

4. **Comment:** Line 110: When you talk about the percentage of children who received an antimalarial falling into different treatment categories, why use receipt of antimalarials in particular? Is this just a way to say children with malaria that were treated or is receipt of antimalarial used as a proxy for having malaria? If so, how are you accounting for over-prescription of antimalarials?
 - **Response:** Receipt of antimalarials was used as a proxy for malaria treatment due to limitations in confirming malaria diagnoses in survey data. While we acknowledge the potential for over-prescription and addressed this concern in the limitations section Lines 258-263 [in the current version, this is in Lines 269-274]. We have also now clarified the limitation of over-prescription within the previous sentence. Unless there is a specific difference in delay of treatment between confirmed malaria vs other febrile illnesses, we believe the challenge of over-prescription does not affect the validity of the proportion delayed for the treatment for malaria

5. **Comment:** Lines 223-229: The authors make a number of recommendations to reduce delayed treatment. However, they provide no citations to suggest that these recommendations will lead to the best health outcomes. Are these meant to be hypotheses about potential solutions to the problem or is there a literature to draw upon when making these recommendations.
 - **Response:** We thank the reviewer for the comment. We have revised and expanded some of the sentences in the section while also referencing relevant literature evidence to support our points.

6. **Comment:** Line 305: What would be considered an invalid response to the question about timing or inconsistent values?
 - **Response:** Thank you for your comment. "Invalid responses" were defined as those where the information provided was implausible. For example, this includes responses where the timing of

treatment was reported as being based on fever occurring within the last 14 days, but the response indicated a treatment time greater than 14 days, or when negative values for time were reported earlier. These invalid responses were only applicable to individual survey answers. On the other hand, “inconsistent values” refer to situations where the entire survey estimate derived from the raw individual data was identified as inconsistent with those reported in the survey report. An example of this would be the Mozambique 2008 survey, which was excluded and tagged implausible, as detailed in the Supplementary Information (SI). We have now updated the methods section and the SI to clearly differentiate these two categories. Specifically, we clarified the terms “invalid” and “inconsistent” by adding text to the SI [before Table S2] to ensure clarity. We hope these revisions help clarify the criteria and address your concerns.

7. **Comment:** Line 314: Why were outliers removed when the aim of the paper was to characterize delays in care seeking across Africa? Was there an investigation into why a country or row of data might be an outlier? Or are the authors using a pure statistical approach to “outlier” detection. If the latter, this may lead to smoother model estimates, but not necessarily more robust and/or better evidence. After all, it’s entirely possible that those ‘outliers’ are those most in need of reliable and timely access to antimalarials.

- **Response:** Thank you for your comment. We acknowledge the confusing description of the text in the referred-to sentence. We have now revised the sentence by removing the part “...and outliers were removed...” as no outliers were removed in covariates, rather only standardisation of covariates was done to reduce collinearity across related terms and improve the stability of coefficient estimates and their interpretation. As for the removal of observation data outliers, it refers to the up-front removal of an entire survey due to data quality concerns. Specifically, the Mozambique 2008 MICs survey was excluded due to inconsistencies between the reported/published values and the derived estimates from the individual-level data. The survey report indicated a 62% delay of >24 hours [23%/37%] in receiving antimalarial drugs for the onset of symptoms. However, when we aggregated the individual data, the delay for >24 hours was estimated at 21%, compared with 75% reported in 2011. This disparity between the reported figures and the individual-level data as well as against subsequent DHS surveys raised quality concerns. Despite our efforts to identify a documented reason for this inconsistency, we found no clear explanation. Thus, we determined that the 2008 survey data from Mozambique contained implausible values and warranted exclusion. This decision was made to ensure that the analysis relied on consistent and reliable data. If the survey had been kept, it would have led to confusion, especially given the discrepancy between reported delays of >24 hours and the lack of data for delays of >48 hours. We hope this clarifies our rationale for omitting the survey.

We now describe the removal of the surveys earlier in methods and have revised the paragraph in question for better clarity.

8. **Comment:** Line 324: Was this the process for estimating severe delays as well? How were the survey questions used to calculate these delays formatted/what did they explicitly ask?

- **Response:** Yes, this was the process for estimating delay using survey responses on treatment timing. An Example of the question from the DHS Phase 7 (2013-2017) asked the question as below and attracts a numeric response starting from 0.

“How long after the fever started did (NAME) first take the ‘specific antimalaria named in a previous question’? In the latter DHS phases after 2018, the question was changed to ask only for artemisinin combination therapy, on the basis that the effective antimalarial is expected to be given, and the response could be 0, 1, 2 ...

We have now modified the text to make this clearer and have added text in the SI section 2.4 describing a sample of how the question is asked.

9. **Comment:** Line 348-49: What is country-year-country ID? The way the model is written in line 343 implies independent spatial and temporal processes. The former is implemented through a CAR prior, while the latter is an AR1 process. The two independent processes imply that the spatial process is constant over time, and the temporal process is constant over space. However, the labelling here seems to imply a separable (or non-separable) process, which is confusing to me. Can the authors clarify this?

- **Response:** We appreciate the reviewer’s comment. We have revised the Methods section to explicitly detail these points and removed ambiguous phrasing. The spatial random effects $S(x_k)$, indexed by country ID, were assigned a BYM prior to model spatial dependencies. Temporal trends were modelled via year-specific random effects γ_j , assigned an AR(1) process γ_j . These components operate independently. To give some additional context:

Clarification of “country-year-country ID”: The term “country-year-country ID” is a phrasing used to denote the grouping factor for the spatial random effects of country ID, which accounts for spatial dependencies across countries. We understand this may be confusing, and we have replaced it with “country ID”. Temporal trends are modelled separately through country year-specific random effects (denoted as γ_j), here referred to as country-year-year ID with an AR(1) process.

Independence of Spatial and Temporal Processes: The model explicitly assumes independent spatial (BYM/CAR) and temporal (AR1) processes, as noted in Equation in line 343. The spatial random effect $S(x_k)$ captures unobserved heterogeneity across countries (constant over time), while γ_j models’ temporal autocorrelation (constant across space). This additive structure avoids space-time interactions, simplifying interpretation while allowing simultaneous estimation of spatial and temporal patterns.

Adjacency Matrix and Implementation: The spatial weights matrix W (binary adjacency) and temporal AR1 prior operate independently. The spatial component does not vary by year, and the temporal component does not vary by country.

10. **Comment:** Line 362: What is country-year-year ID? Same as above.

- **Response:** We hope the previous response and explanation above have clarified this concern.o.

11. **Comment:** (Supplemental Material) Page 11 Figure S3: Were children who were never treated considered? While they would not have data on time to antimalarial, those children would still not be receiving timely care.
- **Response:** Children who were never treated were excluded from the time-to-treatment analysis due to missing data on the timing of treatment. As shown in Figure S3, if a child did not receive any treatment (including antimalarials or other medicines), they were excluded from this specific analysis. However, these children are considered in other models outside this study, particularly in the modelling of treatment-seeking proportions. While they are not directly part of the time-to-treatment analysis, the data on treatment-seeking is used as a covariate in the model. This distinction has been clarified in the SI section before Figure S3 to avoid any confusion.
12. **Comment:** Figure 3: It would be useful if the number of cases treated that were not severely or moderately delayed were added to this figure. Also, the caption states the x-axis includes number of children with antimalarials, but I believe the authors mean the y-axis.
- **Response:** We have now modified Figure 3 to include the number of cases promptly treated (without delay [moderate or severe]) and corrected the caption to reflect that the y-axis represents the number of children receiving antimalarials.
13. **Comment:** Line 59: The reported number of deaths should be updated to the most recent world malaria report (as opposed to the 2022 report).
- **Response:** The reported deaths have been updated to reflect the most recent World Malaria Report citation.
14. **Comment:** Lines 39-40: the statement that prompt treatment within 24 hours could potentially avert half of severe cases needs a citation.
- **Response:** We have added a citation to support the statement regarding prompt treatment averting half of severe cases.
15. **Comment:** Line 48: 'hours' instead of 'hrs'
- **Response:** The term "hrs" has now been replaced with "hours", and this has been done across the manuscript
16. **Comment:** Line 108: 'approximately over a third' is awkward. Why not just state 'an estimated 41%...'
- **Response:** The phrase "approximately over a third" has been revised to be more precise, as "an estimated 41%..."

References mentioned in responses

Dzianach PA, Rumisha SF, Lubinda J, Saddler A, van den Berg M, Gelaw YA, Harris JR, Browne AJ, Sanna F, Rozier JA, Galatas B, Anderson LF, Vargas-Ruiz CA, Cameron E, Gething PW, Weiss DJ. Evaluating COVID-19-Related Disruptions to Effective Malaria Case Management in 2020-2021 and Its Potential Effects on Malaria Burden in Sub-Saharan Africa. *Trop Med Infect Dis*. 2023 Apr 4;8(4):216. doi: 10.3390/tropicalmed8040216.

Rogerson, S.J., Beeson, J.G., Laman, M. et al. Identifying and combating the impacts of COVID-19 on malaria. *BMC Med* 18, 239 (2020).

Norris, M., Klabbers, G., Pembe, A. B., Hanson, C., Baker, U., Aung, K., ... & Beňová, L. (2022). A growing disadvantage of being born in an urban area? Analysing urban–rural disparities in neonatal mortality in 21 African countries with a focus on Tanzania. *BMJ Global Health*, 7(1), e007544.

Lungu EA, Guda Obse A, Darker C, et al. What influences where they seek care? Caregivers' preferences for under-five child healthcare services in urban slums of Malawi: a discrete choice experiment. *PLoS One* 2018; 13.

Matthews Z, Channon A, Neal S, et al. Examining the "urban advantage" in maternal health care in developing countries. *PLoS Med* 2010; 7.

Kimani-Murage, E. W., Fotso, J. C., Egondi, T., Abuya, B., Elungata, P., Ziraba, A. K., ... & Madise, N. (2014). Trends in childhood mortality in Kenya: the urban advantage has seemingly been wiped out. *Health & place*, 29, 95-103.

Macharia, P. M., Beňová, L., Pinchoff, J., Semaan, A., Pembe, A. B., Christou, A., & Hanson, C. (2023). Neonatal and perinatal mortality in the urban continuum: a geospatial analysis of the household survey, satellite imagery and travel time data in Tanzania. *BMJ global health*, 8(4), e011253.